

# Intelligent educational systems based on adaptive learning algorithms and multimodal behavior modeling

Yuwei Li[1] and Botao Lu[2]

[1] College of Physical Education and Health, Hubei Business College, Wuhan, China
[2] Faculty of Artificial Intelligence in Education, Central China Normal University, Wuhan, China

## ABSTRACT

With the rapid advancement of artificial intelligence, the demand for personalized and adaptive learning has driven the development of intelligent educational systems. This article proposes a novel adaptive learning-driven architecture that combines multimodal behavioral modeling and personalized educational resource recommendation. Specifically, we introduce a multimodal fusion (MMF) algorithm to extract and integrate heterogeneous learning behavior data—including text, images, and interaction logs—*via* stacked denoising autoencoders and Restricted Boltzmann Machines. We further design an adaptive learning (AL) module that constructs a student-resource interaction graph and dynamically recommends learning materials using a graph-enhanced contrastive learning strategy and a dual-MLP-based enhancement mechanism. Extensive experiments on the Students' Academic Performance Dataset demonstrate that our method significantly reduces prediction error (mean absolute error (MAE) = 0.01, mean squared error (MSE) = 0.0053) and achieves high precision (95.3%) and recall (96.7%). Ablation studies and benchmark comparisons validate the effectiveness and generalization ability of both MMF and AL. The system exhibits strong scalability, real-time responsiveness, and high user satisfaction, offering a robust technical foundation for next-generation AI-powered educational platforms.

## INTRODUCTION

The conventional education system has fallen short of fulfilling the diverse requirements and aspirations of contemporary students, especially in terms of personalization, efficiency and flexibility. As artificial intelligence technology advances, particularly with significant breakthroughs in large models, it has ushered in unprecedented opportunities for innovation and transformation in the realm of education. Intelligent education provides a new direction for education reform with its unique advantages, such as intelligent tutoring, assessment, and providing diverse learning resources. Intelligent innovative education can carry out intelligent tutoring and evaluation to provide more accurate and efficient teaching services (*Tan, 2023*; *Bi et al., 2025*; *Alrakhawi, Jamiat & Abu-Naser, 2023*). At the same time, it can also provide diverse learning resources and tools to enrich students'

Corresponding author
Botao Lu,
lubotao2024@mails.ccnu.edu.cn

learning content and improve their learning experience. Research on intelligent education has good theoretical and application value (*Iatrellis et al., 2023*; *Sharif & Uckelmann, 2024*).

At present, the research of intelligent education is indeed faced with a series of core problems. Firstly, intelligent education algorithms encounter significant challenges in providing personalized learning suggestions and tutoring for students by data analysis and processing. The existing big model algorithms are incompetent when dealing with the complex situations that may occur in education, and it is difficult to accurately capture and respond to the unique needs of each student (*Shu & Gu, 2023*; *Alzubi et al., 2025*). Secondly, current intelligent education algorithms, especially deep learning-based algorithms, generally lack interpretability. This means that the decision-making process of the algorithm is difficult to understand and trust by human educators, students or parents, which greatly limits the wide application of intelligent education algorithms. Moreover, education is a highly complex and variable field, and students' learning needs, learning styles, and learning progress vary greatly due to individual differences (*Liu & Yu, 2023*). This requires that intelligent education algorithms must have a high degree of adaptability and flexibility, which can dynamically adjust suggestions and recommendations according to real-time situations and feedback. However, many current algorithms still have obvious shortcomings in this aspect (*Ilieva et al., 2023*; *Al Ka'bi, 2023*). Finally, the research of intelligent educational algorithms demands the computer science, education, psychology, statistics and other disciplines. How to effectively integrate the theories and methods of these disciplines to jointly promote the innovation and development of intelligent education algorithms is an important challenge faced by current intelligent education research. In summary, intelligent education algorithms have challenges in personalized learning suggestions, interpretability, adaptability and flexibility, and interdisciplinary integration (*Asitah et al., 2023*).

To solve these problems, many researchers continue to explore new algorithms, and strengthen interdisciplinary cooperation and communication to promote the continuous progress of intelligent education. *He et al. (2019)* presented a novel algorithm for personalized e-learning service recommendations, with the objective of enhancing the accuracy of tailored learning program recommendations that had previously suffered from low precision. *Zagulova et al. (2019)* investigated a personalized learning resource recommendation model that leverages mobile learning terminals (*Okubo et al., 2017*) as its foundation. *Morsy & Karypis (2019)* introduced a cumulative regression aimed at forecasting grades for the subsequent semester. This model captures a student's knowledge progression through a sequential regression approach (*Qiu et al., 2016*). *Qi et al. (2018)* employed contextual Bandit learning, leveraging implicit feedback by integrating it as a crucial aspect of users' result verification and relevance assessments. *Song et al. (2019)* proposed a recommender system called Ekar, which combines knowledge graphs (*Zhao et al., 2019*) to generate recommendations by finding the path of users to teaching resources. *Wu et al. (2015)* proposed the fuzzy cognitive diagnosis framework for the cognitive modeling of examinees on objective and subjective questions. Furthermore, several studies (*Tang et al., 2019*; *Wang et al., 2019*, *2021*; *Liu et al., 2019*) have formulated

the resource recommendation process using knowledge graphs and quantified it through Markov decision processes, yielding satisfactory outcomes.

Indeed, the above studies have significant value in the exploration of intelligent education, but they often fail to think about the personalization of students, disconnecting between students' learning characteristics and educational resources. To make up for this deficiency, we suggest an intelligent innovative education method upon an adaptive learning algorithm, which aims to build an education ecology. The algorithm can deeply analyze students' learning data, including their learning progress, accuracy rate, learning time, *etc.*, to comprehensively understand students' learning characteristics. Based on these analysis results, the algorithm can intelligently recommend learning resources that best meet students' needs, including teaching videos, online exercises, and extended reading. In addition, the algorithm can also adjust the difficulty and progress of teaching content in real-time according to students' learning performance to ensure that students can learn at a pace that is most suitable for them. The adaptive learning algorithm will deeply analyze the individual differences in students' learning styles, ability levels, interests and preferences, and dynamically adjust the educational content, difficulty and progress according to this information. In this way, educational resources can more accurately match the actual needs of students, and improve learning efficiency and effectiveness. At the same time, this intelligent education ecology will also focus on cultivating students' key abilities such as innovation ability, critical thinking and collaboration spirit, to adapt to the demand for talents in the future society. Through the introduction of diversified educational resources and activities, students will be provided with a richer and more diverse learning experience, which will stimulate their interest and motivation in learning.

## RELATED WORKS

The intelligent education method based on an adaptive learning algorithm is a method that uses advanced algorithm technology to optimize the education process to meet students' personalized learning needs.

First of all, scholars have carried out many studies around the analysis of students' learning behavior. *Li et al. (2022)* believe that traditional learning behavior features have limitations in predicting learning effects. They proposed that the learning sequence of learners can better reveal the behavior path and cognitive process. Therefore, they adopted the clustering analysis method to conduct an in-depth study on the learning effect based on multiple action sequences. *Shoaib et al. (2024)* were concentrate on the law of students' learning behavior and constructed a prediction model of students' learning outcomes by carefully selecting effective early warning indicators and combining Bayes theory. *Pan & Yang (2023)* suggested a knowledge point set model upon a Bayesian network. According to the weight and correlation degree of knowledge points, they constructed a model that can early warn learners' status and learning performance. *Carella & Colombo (2024)* analyzed the online learning data of 2,786 students for up to 3 years, and established a prediction model between learning status and learning behavior, thereby revealing the correlation between them. *Navarro et al. (2021)* adopted a more comprehensive approach.

They used a variety of learning behavior data such as learning time, forum interaction, assignment grade, and final grade, and combined with complex data such as students' background information and admission grade, and constructed a learning risk prediction model through data cleaning. The rate of the model in predicting students' failure risk is as high as 80%. *Shu & Gu (2023)* started with non-traditional data in the learning process, analyzed the learners' facial features such as eye movements and changes in expression, used unimodal and multimodal features to evaluate the concentration of learning, and further explored the relationship between attention level and students' learning status.

Subsequently, based on the in-depth analysis of learning behavior, many educational resource recommendation systems have emerged. These systems aim to optimize students' learning experience and improve the accuracy of resource acquisition. *Chen et al. (2023)* developed a comprehensive online lessons recommendation, which innovatively combined data mining and graph theory technology, and provided practical solutions for students to choose various courses through accurate course prediction methods. The research of *Ng & Linn (2017)* focused on the course priorities of students and proposed a recommendation method called CrsRecs. This method analyzes students' priorities for courses and combines these valuable data to provide personalized course recommendation services for students. *Wang (2022)* proposed a hybrid recommendation strategy combining collaborative filtering and XGBoost model. This strategy improves the accuracy and diversity of recommendations by integrating the advantages of multiple algorithms. *Wang & Li (2021)* designed meta-paths containing semantic relations in a residual connection-based graph neural network to guide user preferences. The attention mechanism is combined for reinforcement, and finally, the regularized matrix factorization is used for prediction. To cope with the problem of the low density of course interaction, *Zhang et al. (2023)* proposed to introduce a knowledge graph as a solution. They take advantage of the rich information and strong associations of knowledge graphs *Sheng et al. (2020)* proposed a recommendation model upon an attention meta-path. This model provides more accurate course recommendations for students by constructing multiple relationships between students and courses and combining carefully designed context-based metapaths. *Tang (2022)* adopted the Apriori classification algorithm and developed a personalized course recommendation system based on this algorithm. By analyzing students' learning behaviors and preferences, the system uses the Apriori algorithm to mine the association rules between courses, to recommend appropriate courses for students.

## MATERIALS AND METHODS

To build an education ecology that meets the needs of the future society, we propose an intelligent education method based on an adaptive learning algorithm. By proposing a multimodal integrated student learning behavior analysis method (multimodal fusion (MMF)) and an adaptive learning-driven student personalized resource recommendation method (adaptive learning (AL)), a fast and efficient learning path was constructed for students and the learning efficiency was improved.

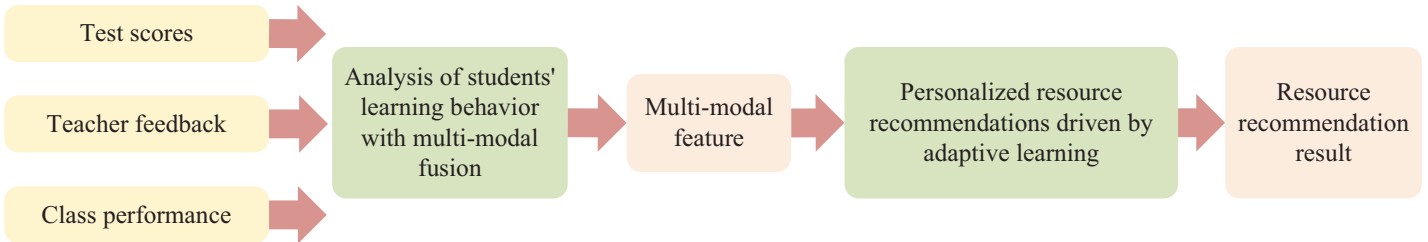

**Figure 1 The framework for environmental sustainability assessment based on accounting information audit.**

The technology roadmap is shown in Fig. 1. Our model introduces a multi-modal integrated student learning behavior analysis framework that seamlessly amalgamates data from diverse learning platforms and tools. This encompasses academic performance metrics, online interaction patterns, homework completion rates, and learning duration logs. By integrating these multifaceted data streams, the model comprehensively captures the nuanced learning behaviors and habits of students. Through this holistic approach, we gain deep insights into individual learning needs, preferences, styles, and progress trajectories. These insights serve as a robust foundation for recommending personalized educational resources tailored to each student's unique profile.

Furthermore, the model incorporates a student personalized resource recommendation system powered by adaptive learning mechanisms. Based on the nuanced analysis of student learning behaviors, the system intelligently curates educational resources that align with each student's learning style and progress level. Leveraging an adaptive learning algorithm, the system dynamically adjusts and refines its recommendation strategy over time. This ensures that every student receives the most pertinent and effective learning materials, thereby enhancing learning efficiency and fostering a greater interest in the educational content. By combining the multi-modal integrated student learning behavior analysis method and the adaptive learning-driven student personalized resource recommendation system, we aim to create an intelligent education environment, so that each student can obtain the best learning experience and learning outcomes based on adapting to their unique needs and learning styles. This not only helps to cultivate students' autonomous learning ability and innovative thinking but also cultivates more talents with diverse skills for future social development.

Figure 1 The framework for intelligent innovative education method upon adaptive learning.

## The analysis method of student learning behavior upon multimodal integration

To boost the recognition ability of students' learning status, we propose a multimodal fusion student learning behavior analysis method to deeply mine students' learning internal information from multiple dimensions.

We aim to structure accurate behavior portraits by the acquisition of multivariate data, and the extraction and fusion strategy of text and image features. Firstly, statistical analysis

techniques are used to extract objective labels from the original data. These labels are then refined and standardized by data preprocessing techniques. As the core part of this method, data fusion involves integrating data from different sources, such as student text records, photo materials, and voice data, to produce richer and deeper information than using these data alone.

In the application scenario of student behavior analysis, we make full use of multimodal data generated by students, including but not limited to students' performance in class $C_p$, test scores $T_s$, teacher feedback $T_f$, etc. These data sets are denoted as $V = \{T_s, T_f, C_p\}$, which includes text data (*e.g.*, assignments, notes, *etc.*), image data (*e.g.*, diagrams in lab reports, class photos, *etc.*), and possibly voice data (*e.g.*, class discussions, voice assignments, *etc.*).

Firstly, we extract the multi-modal features by the stacked denoising autoencoder to learn the input feature representation. Second, each modal feature is represented by a stacked denoising autoencoder, as shown in the formula:

$$Z^i (i = 1, 2, \ldots, n). \tag{1}$$

Model the relationship and obtain a unified feature S.

Finally, the modality is represented as $Z^i$, which is utilized as input for an advanced network, modeling and learning the connection between them. Then, we use the Restricted Boltzmann Machine (RBM) to examine the correlation among various modal features to construct a shared representation layer through modeling. We set the energy function as follows:

$$E(v, h) = -\sum_i a_i v_i - \sum_j b_j h_j - \sum_{i,j} v_i w_{ij} h_j \tag{2}$$

where v refers to the measured variable, h denotes the hidden variable, w is the weight, and a and b are the bias terms. Given v and h, the joint probability distribution can be expressed as follows:

$$P(v, h) = \frac{1}{\sum_{v,h} \exp(-E(v,h))} \exp(-E(v,h)) \tag{3}$$

where we are more interested in the conditional probability of h given the observed variable v, or the conditional probability of the observed variable v given the hidden variable h. These conditional probability distributions can be computed using the energy function and the partition function. At the same time, we set different dimensions of student information to be based on time, and use long short term memory (LSTM) to analyze student behavior text and other related information. The framework is shown in Fig. 2.

In the analysis of students' learning behaviors through multimodal fusion, LSTM plays a pivotal role by leveraging its unique advantages in processing temporal data. Students' learning behaviors exhibit dynamic continuity, such as fluctuations in attention over time and changes in interaction frequency. LSTM effectively captures these long-term dependencies through its gating mechanisms and memory cells, thereby avoiding the issue

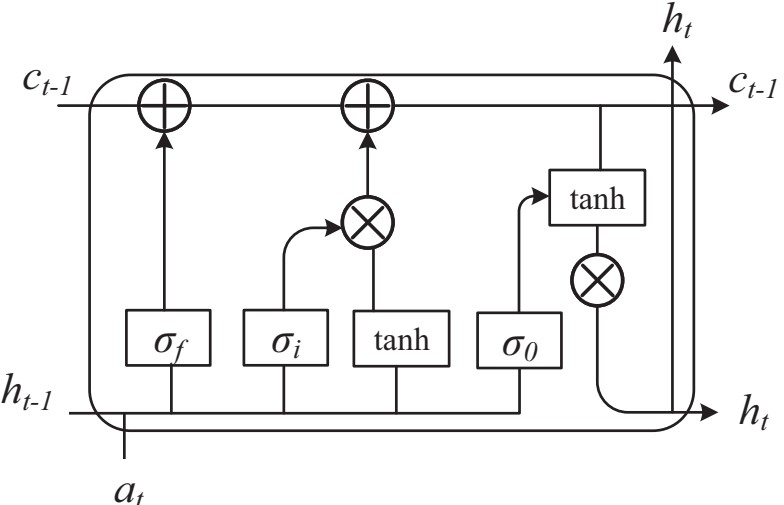

**Figure 2  The framework of LSTM.**

of information loss common in traditional models. In terms of multimodal fusion, LSTM can flexibly handle temporal features from diverse modalities like video, audio, and text. It can either uniformly model the data after early fusion of multimodal inputs or separately process each modality and later integrate high-level semantic features, fully utilizing the complementarity among modalities to enhance the robustness of the analysis. Additionally, LSTM significantly improves the accuracy of behavior classification and prediction. It can precisely classify students' states (*e.g.*, focused or distracted) and predict future behaviors or learning outcomes based on historical behavior sequences, providing a basis for personalized teaching. When combined with attention mechanisms, LSTM can also focus on key modalities and time periods, enhancing the model's interpretability.

## Personalized resource recommendation method for students driven by adaptive learning

Based on the previous chapters, we have successfully collected and integrated the multi-modal features exhibited. These features cover a variety of data, including learning behavior, learning progress, interactive feedback and other dimensions. To further optimize students' learning experience and improve learning outcomes, we innovatively propose a personalized resource recommendation method for students driven by adaptive learning. The core of this method is to understand students' learning styles, ability levels, and interests and preferences through in-depth analysis of students' multi-modal characteristics, to provide tailored learning resource recommendations for them. Our recommendation system can track learning status in real-time, and dynamically adjust the recommendation strategy according to students' real-time feedback to ensure that the recommended content not only meets students' current needs but also effectively promotes their learning progress.

Given a student-resource interaction graph G = {V, E} and an interaction matrix R, this article proposes a student-resource interaction graph G = {V, E}. We first constructed this graph based on the actual interaction behaviors of the students. To explore its potential complementary relationship network from the resource attribute level, we further construct a graph Gc. Then, we employ the cosine function to calculate the feature similarity Sij between any two nodes vij and v in the graph. The cosine function is used here as a metric to quantify the similarity of nodes in the feature space. Based on the above-calculated feature similarities between node pairs, we construct a feature similarity matrix $S \in \mathbb{R}$. Each element Sij of the matrix represents the feature similarity between node vi and vj. For each node vi in the graph, we choose the k nodes with the highest similarity to it based on its feature similarity matrix S. We build edges between the selected k nodes and the original node vi to form a new edge set Ec. From this, we obtain the complementary graph Gc = {Vc, Ec}. In this way, the complementary graph Gc can enhance the feature diversity of the original graph G. In contrastive learning, this increase in diversity provides the algorithm with more data variations and content differences. By combining the information of the original graph Gc and the complementary graph G, we can more comprehensively understand the complex relationship in the student-resource interaction network, and make more in-depth analysis and prediction based on it.

To mitigate the effects of bias and interaction noise, we introduce an adaptive enhancement module into it. Given G and Gc, we employ two MLPs to estimate the weights of nodes and edges, which determine their retention probability in the subsequent enhanced views. The larger the weight, the higher the retention probability. MLP can fit nonlinear and complex relationships, capture the high-order interactions between different attributes of nodes and edges more comprehensively, and calculate their roles in each layer of the graph more finely, as shown in Fig. 3. The formula is as follows:

$$\beta_{v_i} = \mathcal{M}(x_i) \tag{4}$$
$$\beta_{e_{ij}} = \mathcal{M}([x_i, \, x_j]) \tag{5}$$

where xi and xj denote node vi and the convolutional layer embedding representation of node vj respectively, the formula of $\mathcal{M}$ is as follows:

$$y = softmax(W_1 \cdot g(W_2 \cdot x + b_2) + b_1) \tag{6}$$

where g is the activation and W is the trainable matrix.

## Data preprocessing

To ensure data compatibility with the proposed multimodal fusion and adaptive learning modules, the raw dataset underwent the following structured preprocessing procedures:

**Handling missing values:** Missing entries within numeric attributes such as "raised hands," "resource usage," and "discussion participation" were imputed using the mean value calculated over the training partition. For categorical variables (*e.g.*, gender, nationality, and school satisfaction), mode imputation was performed independently per feature.

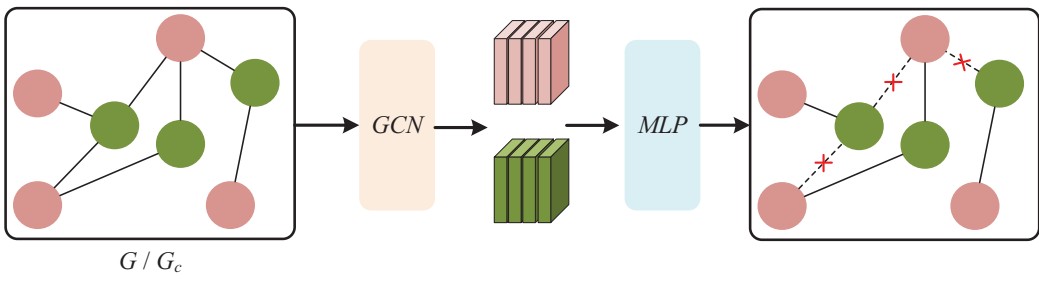

**Figure 3 Bayesian network.**

**Numerical feature normalization:** All continuous variables were normalized to a [0, 1] interval *via* min-max scaling.

**Categorical feature encoding:** Categorical variables were encoded using one-hot encoding. For instance, the "StageID" feature comprising three categories (Lower, Middle, Upper) was transformed into a 3-dimensional binary vector, enabling direct input into the model without introducing ordinal bias.

**Outlier mitigation:** To address extreme values in behavioral metrics (*e.g.*, unusually long session durations), a z-score normalization was first applied. Values exceeding a threshold of $|z| > 3$ were treated as outliers. These outliers were either capped at the boundary value or removed depending on their impact on the distributional skewness.

**Temporal sequence structuring:** Behavioral data collected over two academic semesters were restructured into fixed-length temporal sequences. Each student's interaction history was aligned chronologically to construct time-series input vectors suitable for LSTM-based modeling. Sequences shorter than the defined window length were zero-padded, while longer sequences were truncated from the beginning to retain the most recent interactions.

**Multimodal data formatting:** *Textual data* were tokenized and embedded using a pretrained BERT encoder, producing dense 768-dimensional vectors; *Visual data* were resized to 224 × 224 pixels and encoded *via* stacked denoising autoencoders; *Log features* were aggregated into fixed-length numerical vectors.

## Computing infrastructure

During the training phase, we maximized the efficiency of a robust hardware configuration, including an i7-12700 processor and four Nvidia RTX 3070 Super graphics processing units (GPUs). To facilitate the training process, we chose Tensorflow as our deep learning framework of choice and fine-tuned its configurations to align precisely with the specified training parameters outlined in Table 1. The selection of hyperparameters aims to achieve better convergence, stability, and performance during the model training process. The learning rate is a crucial parameter that controls the step size of each model update. Choosing $6 \times 10^{-4}$ is because it represents a moderate value that enables the model to converge within a reasonable time frame while avoiding oscillations or non-convergence

**Table 1 Model training settings.**

| Parameters | value |
| --- | --- |
| Learning rate | $6 \times 10^{-4}$ |
| Epoch | 48 |
| Batch-size | 80 |
| Decay | 0.88 |
| Gradient descent method | MAE |

caused by excessively large step sizes. The batch-size determines the number of samples used for each gradient update, and 80 is a well-balanced choice. Decay typically refers to the learning rate decay coefficient, which is employed to gradually reduce the learning rate during the training process. Stochastic gradient descent (SGD) is a classic optimization algorithm that updates model parameters by using a mini-batch of samples each time.

## Evaluation method

To rigorously assess the performance of the proposed MMF algorithm, we conducted comparative analysis against four state-of-the-art multimodal fusion methods: 3DLIM (*Luo et al., 2023*), Multimodal Information Fusion Model (MIFM) (*Song et al., 2023*), multimodal sentiment analysis (MSA) (*Gandhi et al., 2023*), and Interactive Graph-based Multimodal Joint Collaborative Filtering (IGMJCF) (*Zhu et al., 2023*). The Three-Dimensional Layered Interest Modeling (3DLIM) model utilizes a structured interest layer framework to capture learners' behavioral dynamics across multiple dimensions, but it primarily emphasizes temporal segmentation rather than deep feature interactions across modalities. MIFM integrates textual and visual content representations using a hierarchical encoder-decoder structure; however, its fusion granularity remains limited to modality-level alignment rather than token- or frame-level representations. The MSA model introduces cross-attention mechanisms to align semantic cues between modalities, yet it assumes static attention weights and is less adaptable to rapidly shifting learner behaviors. In contrast, IGMJCF constructs heterogeneous graphs to model collaborative signals and multimodal interactions, but it does not explicitly incorporate unsupervised deep encoders for latent feature learning. These baseline models represent diverse architectural paradigms in multimodal learning and jointly serve as strong reference points for evaluating the representational depth, flexibility, and generalization ability of the MMF algorithm.

To fully evaluate AL and MMF, we use mean absolute error (MAE), mean square error (MSE), recall (R) and precision (P) as evaluation indicators, which are calculated as follows:

$$MAE = \frac{1}{n} \sum_{i=1}^{n} | (\gamma_i - \delta_i) | \tag{7}$$

$$MSE = \frac{1}{n} \sum_{i=1}^{n} (\gamma_i - \delta_i)^2 \tag{8}$$

$$P = \frac{TP}{TP + FP} \tag{9}$$

$$R = \frac{TP}{TP + FN} \tag{10}$$

where $\gamma$ refers to the true value, $\delta$ denotes the predicted value. TP represents the true example. FP denotes the false positive. FN refers to the false negative example.

# EXPERIMENT AND ANALYSIS

## Dataset

We have fully tested the intelligent innovative education approach based on adaptive learning algorithms, using the Students' Academic Performance Dataset (https://www.kaggle.com/code/kianwee/eda-students-academic-performance-dataset, DOI: 10.1109/ACCAI61061.2024.10602439). The dataset is sourced from the LMS Kalboard 360, where educational data was collected adhering to the xAPI standard. As part of the TLA project, it encompasses 480 student records, each characterized by 16 unique attributes grouped into three main categories: demographic, academic, and behavioral. These attributes include details such as gender, nationality, current educational level, grade, student section, frequency of hand-raising in class, patterns of resource usage, responses to parental surveys, and overall satisfaction with the school. The dataset maintains a gender balance, consisting of 305 records for male students and 175 for female students. The data covers a period of two semesters, with 245 records from the first semester and 235 from the second.

## Ablation experiments

For the intelligent education method based on an adaptive learning algorithm, we conduct systematic ablation experiments on the core modules to understand the specific contributions of each module to the overall model performance in detail. Specifically, we focus on the combination effect of AL module and MMF module and deeply explore the influence of each module and each other on model performance by reconfiguring the combination of these two modules. Through this series of experiments, we successfully obtained quantitative data on the role of the two modules in the intelligent education method. These results have been shown in Fig. 4 and Table 2 in detail.

First, we conducted a series of in-depth and independent performance tests designed to evaluate the AL and MMF. Based on the baseline model, we take a systematic approach to introduce AL and MMF techniques one by one, to observe their specific impact on model in detail. The experimental results show that when we introduce AL alone into the baseline model, all key evaluation metrics of the model have achieved significant improvement. The MAE and MSE decrease by 0.040 and 0.0149, respectively. Similarly, when we choose to integrate MMF into the baseline, the performance is also significantly enhanced. This change is particularly noticeable in the P and R values, which are increased by 0.04 and 0.036, respectively, indicating the important role of MMF in enhancing the prediction accuracy and comprehensiveness of the model. Furthermore, to explore the joint effect of AL and MMF, we try to embed these two techniques in the baseline model

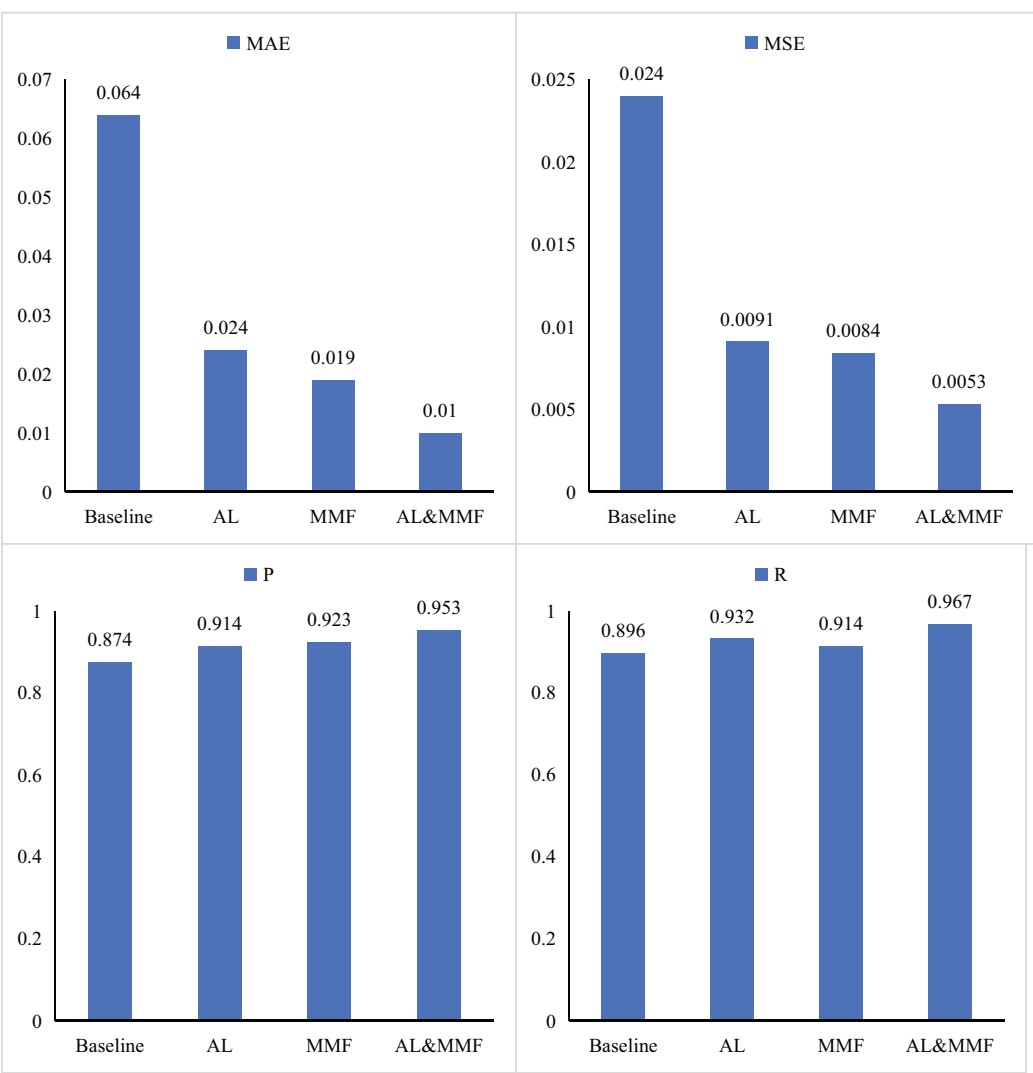

**Figure 4  Ablation experiments results of AL and MMF.**

**Table 2  Ablation experiments.**

| AL | MMF | MAE | MSE | *P* | *R* |
|---|---|---|---|---|---|
| Baseline | | 0.064 | 0.024 | 0.874 | 0.896 |
| O | | 0.024 | 0.0091 | 0.914 | 0.932 |
| | O | 0.019 | 0.0084 | 0.923 | 0.914 |
| O | O | 0.010 | 0.0053 | 0.953 | 0.967 |

simultaneously. Experimental results show that this strategy can further reduce MAE and MSE, while improving P and R values, thereby comprehensively optimizing the performance of the model. This finding not only proves the advantages of AL and MMF respectively, but also reveals the powerful synergistic effect that can be generated when they are combined.

For AL, it allows the model to dynamically select the most informative data points to learn from during training. In this way, the model can make more efficient use of limited resources and focus its attention on those data. Therefore, the significant reduction of MAE and MSE after the introduction of AL into the baseline model is a direct reflection of this advantage. On the other hand, MMF technology provides a richer and more comprehensive data input for the model by integrating information from different sources. After integrating MMF into the baseline model, the model can make full use of the complementarity between various modalities, to capture the intrinsic laws and characteristics of the data more accurately. This comprehensive information input makes the model more accurate and comprehensive in the prediction, so the significant improvement of P and R values is the effect brought by MMF technology.

## Comparison with other methods

First, we conduct an exhaustive performance test of the MMF algorithm and compare its results with several industry-leading algorithms, including 3DLIM (*Luo et al., 2023*), MIFM (*Song et al., 2023*), MSA (*Gandhi et al., 2023*), and IGMJCF (*Zhu et al., 2023*). The test results demonstrate that the MMF performs extremely well, as shown in Fig. 5. The MAE of MMF is only 0.019, which is much lower than other comparison algorithms. Similarly, its MSE is only 0.0084. In addition, the *P* value and R value of MMF reach 0.923 and 0.914, respectively, which shows the powerful ability of multi-modal fusion features. Compared with the 3DLIM algorithm, the MAE value of MMF is reduced by 0.035, and the MSE value is also reduced by 0.0034, which fully demonstrates the advantages of MMF in error control. Compared with the MIFM algorithm, the *P* value and R value of MMF are increased by 4% and 0.8%, respectively, which further proves the improvement of MMF in feature representation performance. Compared with MSA algorithm, MMF shows better performance in MAE and MSE, while maintaining the same advantages in P value and R value. These comparative data fully demonstrate the excellence and competitiveness of MMF algorithm in performance. By further analyzing the performance of the MMF algorithm, we can deeply explore the reasons and advantages behind it. Firstly, the MMF algorithm adopts an advanced multi-modal fusion strategy, which can make full use of the complementarity between different modal data. By intelligently and effectively fusing the information from different modalities, the MMF algorithm can extract more comprehensive and accurate features, thereby improving the performance of the algorithm. Second, the MMF algorithm adopts efficient algorithm optimization techniques in data processing. By optimizing the algorithm structure and parameter Settings, MMF can achieve fast processing of large-scale data while ensuring computational efficiency. In addition, MMF algorithm has good interpretability and adjustability. By adjusting the parameters and model structure of the algorithm, users can customize and optimize the performance of the algorithm according to specific application scenarios and requirements. This flexibility enables MMF algorithm to better adapt to different application environments and meet the needs of different users.

To fully dissect the performance excellence of AL and its potentially great value, we conducted an exhaustive series of performance evaluations. To this end, we selected AITel

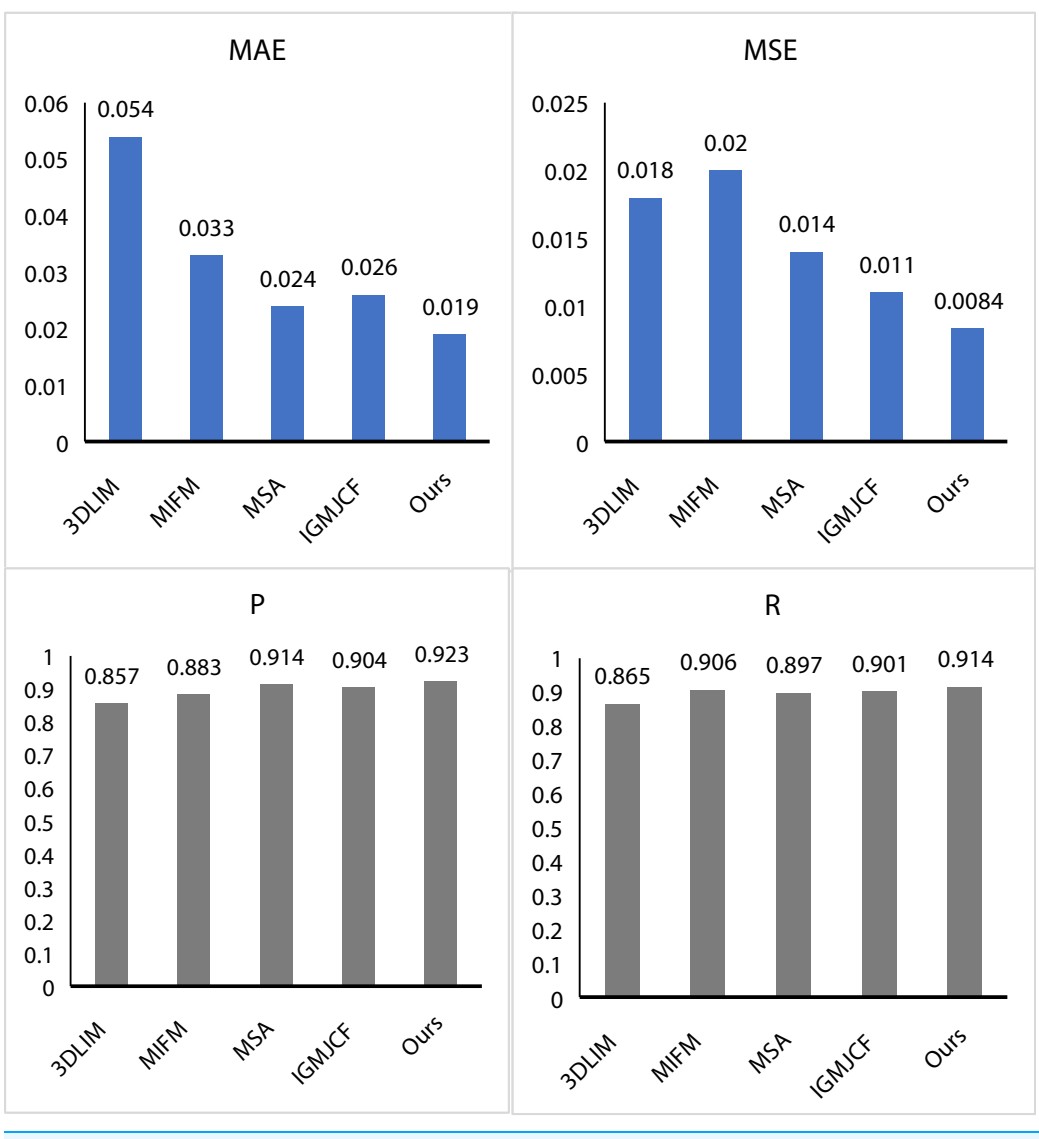

**Figure 5  Comparison MMF with other methods.**

(*Ahmed, Kumar & Kim, 2023*), ReFRS (*Imran et al., 2023*), LightFR (*Zhang et al., 2023*), and TAYN (*Li et al., 2023*) as benchmarks to more accurately measure the performance of AL. As depicted in Fig. 6, AL demonstrates remarkable advantages across several key performance metrics. Specifically, AL attains an MAE of 0.01, an MSE of 0.0053, a *P* value of 0.953, and an R value of 0.967. When compared to AITel, AL exhibits a reduction in MAE by 0.01 and in MSE by 0.0046, clearly showcasing its exceptional error control capabilities. In contrast to ReFRS, AL achieves a 2.7% increase in the *P* value and a 5.2% increase in the R value, highlighting its enhanced performance in these areas. Similarly, when benchmarked against LightFR, AL records a decrease in MAE by 0.014 and an increase in the *P* value by 1.1%. Finally, in a head-to-head comparison with TAYN, AL consistently leads across all evaluated indicators, solidifying its position as the superior performer with outstanding overall stability and excellence. Compared the detailed data

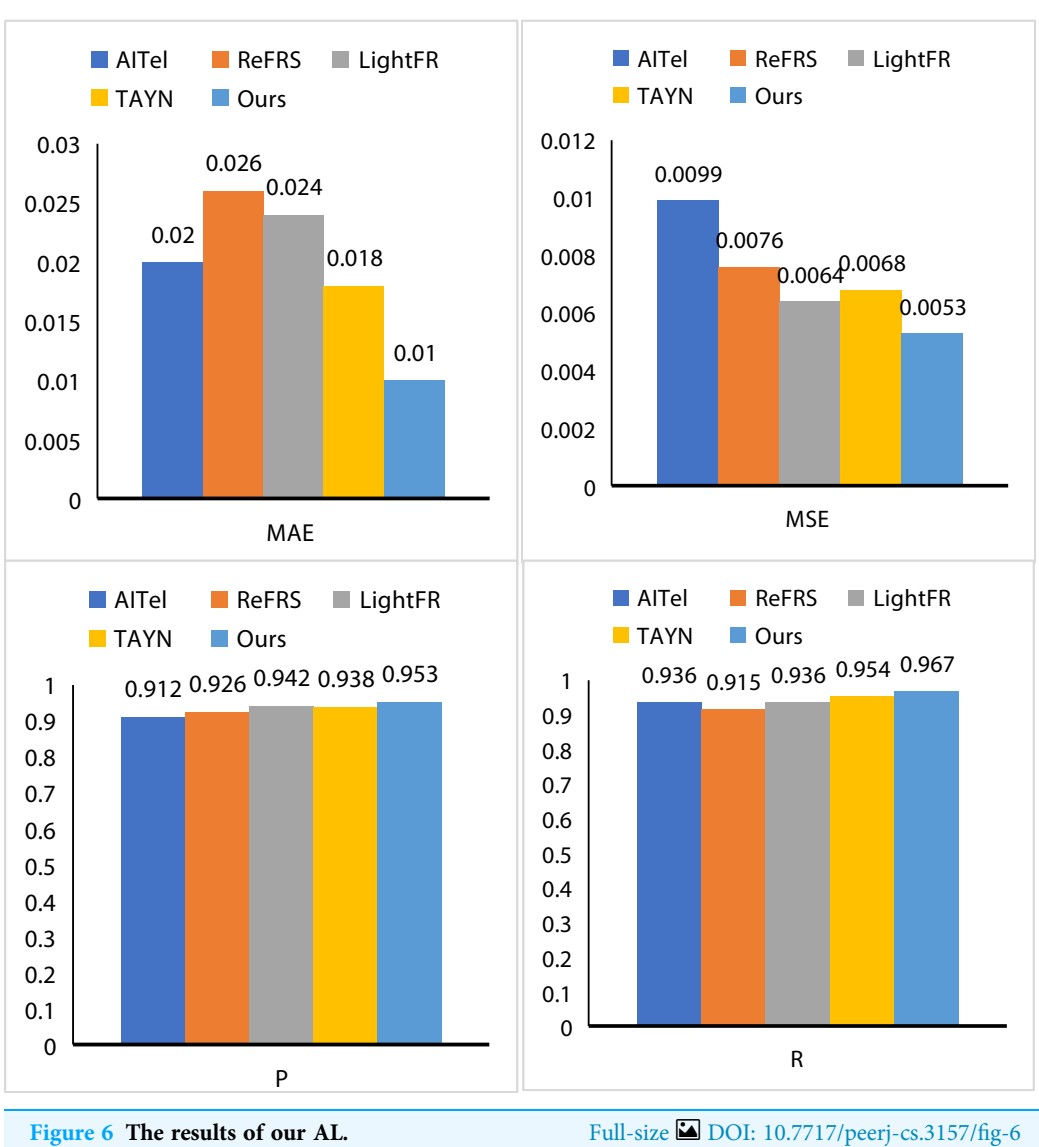

**Figure 6 The results of our AL.**

and charts analysis not only reveals the AL for us in performance excellence but also further excavates its potential value for us to provide powerful data support.

In addition, to ensure the stability of the models, we conduct cross-validation for MMF and AL. We employ the K-fold cross-validation method, dividing the dataset into five subsets of similar sizes. In each iteration, only one subset is used, and this process is repeated five times. The resulting standard deviation of performance for both MMF and AL is 0.012, indicating that the aforementioned experimental results are stable.

## System running test

To simulate the performance of our method in the actual scenario, we specially made artificial adjustments to the samples in the dataset. Specifically, we reintegrate the data set to form different numbers of student sample sets, which contain 10, 100, 500, 1,000 and 3,000 student samples, respectively.

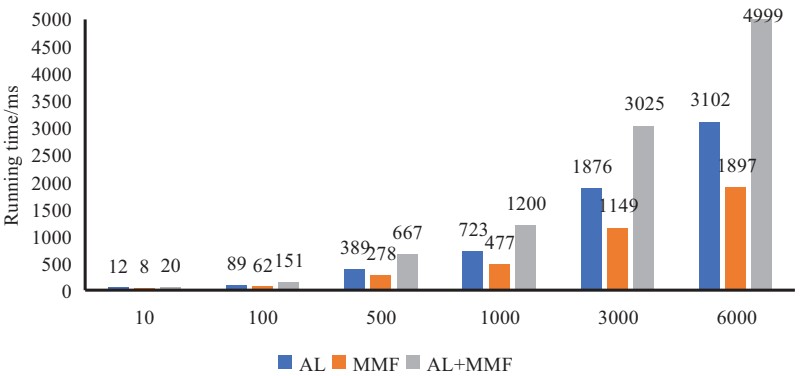

**Figure 7  System response efficiency test.**               

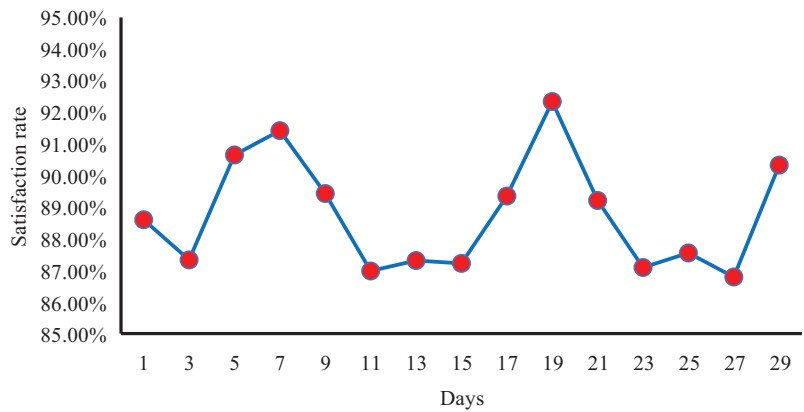

**Figure 8  The student satisfaction rate of our system within a month.**

Firstly, we conducted a running time test to evaluate the efficiency of our method when dealing with different numbers of student samples. By comparing the running time under different sample sets, we can better understand the method on different sizes of data, to provide a reference for scene selection and optimization in practical applications. As illustrated in Fig. 7, we have meticulously documented the operational times of AL, MMF, and the entire model across varying quantities of student samples. Through a comparative analysis, it becomes evident that, even as the concurrent user count rises, the model's response time does not exhibit a doubling trend. This outcome underscores the model's ability to sustain high efficiency when handling extensive datasets and to manage simultaneous requests that may arise in real-world scenarios, thereby ensuring stable and efficient services for users.

Subsequently, to further verify the performance of the whole system in practical applications, we conducted an exhaustive test using a large data set containing 6,000 student samples and paid special attention to the key indicator of student satisfaction rate. This satisfaction rate is obtained through real-time questionnaires administered to students after their experience with the system. As depicted in Fig. 8, following a

month-long continuous statistical analysis of the system, we observed that student satisfaction with the recommendation system consistently stayed above 86%, with the monthly average satisfaction rate reaching an impressive 88.77%. This outstanding outcome unequivocally demonstrates the stability and effectiveness of our system, along with its capacity to cater to students' individualized requirements. We are confident that, through ongoing refinement and enhancement, our recommendation system will continue to elevate user satisfaction and deliver even more precise and beneficial recommendations to the broader student community.

## Discussion

After a series of rigorous and in-depth experiments, we successfully verified the effectiveness of the multimodal fusion student learning behavior analysis method and the adaptive learning-driven student personalized resource recommendation method. These experiments conclude the great value of the intelligent innovative education method based on an adaptive learning algorithm.

Firstly, multimodal fusion technology enables us to integrate student learning data from different channels, including video viewing records, online test scores, learning forum discussions, *etc.*, to comprehensively and accurately capture students' learning behaviors and preferences. This comprehensive data support lays a solid foundation for subsequent personalized resource recommendations.

Secondly, the adaptive learning algorithm can intelligently adjust the content and difficulty of learning resources according to students' learning behavior and feedback, ensuring that each student can obtain a personalized learning experience matching his/her learning progress and ability. This personalized learning method not only improves students' learning efficiency but also stimulates their interest and motivation in learning.

In addition, the intelligent innovative education method is also flexible and extensible. With the continuous progress and update of educational technology, we can continuously integrate new educational ideas and technologies into the system to further improve the performance and user experience of the system. Meanwhile, the system can also be customized and extended according to the specific demands of schools, teachers and students to meet the diverse needs of different educational scenarios.

These methods improve our cognition of students' learning behavior, thus effectively promoting students' learning efficiency and interest. We firmly believe that with the continuous progress and optimization of technology, intelligent and innovative education will play a more important role in the future education field.

## CONCLUSIONS AND LIMITATIONS

This study presents an intelligent educational system that integrates an MMF algorithm and an AL module to support personalized resource recommendation based on heterogeneous student behavior data. Through the combination of stacked denoising autoencoders, Restricted Boltzmann Machines, and graph-based contrastive learning, the proposed method captures fine-grained learning patterns and demonstrates strong performance in both prediction accuracy and generalization. Experimental evaluations

using the Students' Academic Performance Dataset validate the system's effectiveness in terms of reduced prediction error (MAE = 0.01, MSE = 0.0053) and enhanced recommendation precision and recall. Furthermore, the proposed framework exhibits scalability and stability under increasing user loads, achieving a monthly average satisfaction rate of 88.77%.

### Limitations

This study has several limitations that warrant future investigation. First, the dataset used in this work is relatively small in scale and may not fully capture the diversity of real-world educational environments, which limits the generalizability of the model. Second, although the current framework incorporates textual, visual, and behavioral modalities, other types of learner data—such as physiological signals, eye tracking, or affective states—remain unmodeled and may further enhance prediction performance. Third, the model relies on supervised learning techniques that require labeled outcome data (*e.g.*, performance scores), which may not always be available or reliable in practical settings. Fourth, the interpretability of both the MMF and AL modules, especially under the deep learning components, remains limited and requires additional *post-hoc* explanation mechanisms to ensure transparency and trustworthiness for educators and learners. Lastly, while the system is tested on a static dataset, real-world educational platforms typically involve dynamic and evolving user behaviors. Adapting the model for continuous learning or online updates is an important direction for future research.

## ACKNOWLEDGEMENTS

We thank the anonymous reviewers whose comments and suggestions helped to improve the manuscript.

### Funding

The authors received no funding for this work.

### Competing Interests

The authors declare that they have no competing interests.

### Author Contributions

- Yuwei Li conceived and designed the experiments, analyzed the data, performed the computation work, prepared figures and/or tables, and approved the final draft.
- Botao Lu conceived and designed the experiments, performed the experiments, analyzed the data, authored or reviewed drafts of the article, and approved the final draft.

### Data Availability

The Students' Academic Performance Dataset is available at Kaggle: https://www.kaggle.com/code/kianwee/eda-students-academic-performance-dataset.

## Supplemental Information

Supplemental information for this article can be found online at http://dx.doi.org/10.7717/peerj-cs.3157#supplemental-information.

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
