# Peer review of "Intelligent educational systems based on adaptive learning algorithms and multimodal behavior modeling"

_PeerJ Computer Science, doi:10.7717/peerj-cs.3157_

## Round 0.1 · original submission · Major Revisions

Dear authors,

Thank you for submitting your article. Feedback from the reviewers is now available. It is not recommended that your article be published in its current format. However, we strongly recommend that you address the issues raised by the reviewers, especially those related to readability, experimental design and validity, and resubmit your paper after making the necessary changes.

Best wishes,

**Language Note:** The review process has identified that the English language must be improved. PeerJ can provide language editing services - please contact us at [email protected] for pricing (be sure to provide your manuscript number and title). Alternatively, you should make your own arrangements to improve the language quality and provide details in your response letter. – PeerJ Staff

·

Basic reporting

1- Correct Figure 1 Caption: Replace “environmental sustainability assessment” (Line 181) with the correct framework description.
2- Enhance Dataset Validation: Use additional datasets or report cross-validation (e.g., 5-fold) to address generalizability concerns (Line 478).
3- Detail LSTM Integration: Add a paragraph or diagram in Section 3.1 explaining LSTM’s role in MMF (Line 227).
4- Improve Language: Proofread for conciseness and grammar (e.g., Lines 47, 396) or use a professional editing service.
5- Clarify Satisfaction Methodology: Describe how satisfaction rates were measured (Line 437, Figure 8) to support claims.
6- Update References: Include 2024–2025 works to reflect current trends in adaptive learning.

Experimental design

N/A

Validity of the findings

N/A

Reviewer 2 ·

Basic reporting

The manuscript is written in generally clear and professional English, although some sections would benefit from minor linguistic polishing to improve fluency and readability. For example, terms like "intelligent innovation education" are repeated excessively, which slightly affects the narrative coherence.

The structure is well organized and aligns with PeerJ standards. The introduction sets the context and motivation effectively, clearly outlining the gaps in existing educational technologies and the need for adaptive and personalized systems.

The literature review is comprehensive and up-to-date, citing relevant studies and providing a solid theoretical foundation. The figures and tables are well-integrated and generally appropriate, although a clearer caption for Figure 1 would enhance understanding, as the title appears to be mismatched (“Framework for environmental sustainability”).

Experimental design

The manuscript presents a technically sound experimental design. The authors propose an architecture combining a Multimodal Fusion (MMF) algorithm with an Adaptive Learning (AL) module. The description of the algorithms is detailed, with appropriate use of equations and model components such as RBMs, LSTMs, and contrastive learning.

However, the dataset used (Student Academic Performance from LMS Kalboard 360) is relevant, but relatively small (N=480), which limits generalizability.

Details on hyperparameter selection and model tuning are somewhat limited. Table 1 includes some hyperparameters, but no rationale is provided for their selection.

Overall, the design is solid but would benefit from stronger justification of modeling choices and ideally open-sourced code.

Validity of the findings

The findings are promising, showing that the proposed system outperforms multiple baseline models in terms of error reduction and prediction accuracy. The ablation studies provide good insight into the contribution of each module (MMF and AL), and the comparative analysis with other algorithms strengthens the claims.

However, overfitting risk is not discussed, despite the relatively small dataset and complex architecture.

Also, the generalizability of the model to different populations or educational contexts is limited due to the use of a single dataset.

Interpretability is flagged as a limitation, but no concrete solution (e.g., SHAP, attention visualization) is proposed.

Despite these concerns, the conclusions are mostly well supported by the presented data.

Additional comments

The term "intelligent innovation education" is ambiguous and could be better defined or replaced with more conventional terminology.

Figure 1 seems incorrectly titled (mentions “environmental sustainability”) and needs correction.

The limitations section is appreciated and appropriately critical. The acknowledgment of issues like small dataset size, lack of physiological data, and reliance on supervised learning is honest and valuable.

While the methodology and results are convincing, the paper would benefit from:
- Language refinement and terminology clarification.
- Clarification/correction of figure titles and captions.
- A brief justification of hyperparameters and architectural choices.

Reviewer 3 ·

Basic reporting

The paper proposes an intelligent educational system integrating a hybrid recommendation algorithm and learning performance evaluation model. While the theme is highly relevant, especially in personalized education, the manuscript currently lacks technical depth and empirical rigor. Substantial revisions are required.
 The system is described conceptually, but no formal algorithmic or modular decomposition is provided. A schematic diagram accompanied by formal definitions (e.g., input space, learner profile vectors, objective functions) is essential to clarify the internal logic and component interaction.
 The manuscript suffers from grammatical issues and redundancy. For instance, “personalized resources is crucial for learning engagement in modern education systems” → should be “are crucial” and can be rewritten for clarity and conciseness.

Experimental design

The description of the “hybrid recommendation” lacks algorithmic transparency. It is unclear whether collaborative filtering, content-based methods, or matrix factorization techniques are employed. Include pseudocode or mathematical models to detail the hybridization mechanism.
 Although the system claims to offer personalized resource delivery, the evaluation does not use any personalization-specific metrics such as Precision@K, NDCG, or diversity/novelty indices. These are standard in educational recommender systems and must be included.
 The “learning evaluation model” appears to rely on user feedback and performance data, but no concrete modeling (e.g., Bayesian knowledge tracing, IRT models, or regression analysis) is described. This weakens the credibility of adaptive assessment claims.

Validity of the findings

There is no clear description of the dataset used for validation. If synthetic data were used, the generation process should be formalized. If real data were used, the authors should disclose sample size, feature space, and privacy handling strategy.
 The paper does not compare the proposed system against known educational systems or algorithms (e.g., Moodle + ML plugins, Duolingo’s learning path optimizers). Without a baseline, the performance advantages remain unsubstantiated.
 Figures show interface mock-ups, but there is no user journey modeling or human-centered design validation. Conducting a small-scale usability study (even qualitatively) would enhance the system's practical relevance.

Additional comments

The paper proposes an intelligent educational system integrating a hybrid recommendation algorithm and learning performance evaluation model. While the theme is highly relevant, especially in personalized education, the manuscript currently lacks technical depth and empirical rigor. Substantial revisions as mentioned are required.

---

## Round 0.2 · accepted · Accept

Dear Authors,

Thank you for addressing the reviewers' comments. Your manuscript now seems sufficiently improved.

Best wishes,

·

Basic reporting

Acceptable.

Experimental design

Acceptable.

Validity of the findings

Acceptable.

Reviewer 2 ·

Basic reporting

The manuscript is well-written in clear, professional English and adheres to PeerJ’s structural standards. The introduction now effectively contextualizes the research, and the literature review is comprehensive and relevant

Experimental design

The study is rigorous and aligns with PeerJ’s scope. Methods are now described in sufficient detail, and the computing infrastructure is well-documented

Validity of the findings

Results are well-supported, and conclusions align with the evidence. Ablation studies and comparisons with baselines are thorough. Minor notes:

Limitations (Section 5.2): Consider adding a brief discussion on ethical considerations (e.g., data privacy in multimodal behavior tracking).